# Comprehensive assembly of novel transcripts from unmapped human RNA-Seq data and their association with cancer

Majid Kazemian, Min Ren, Jian-Xin Lin, Wei Liao, Rosanne Spolski & Warren J Leonard[*]

## Abstract

Crucial parts of the genome including genes encoding microRNAs and noncoding RNAs went unnoticed for years, and even now, despite extensive annotation and assembly of the human genome, RNA-sequencing continues to yield millions of unmappable and thus uncharacterized reads. Here, we examined > 300 billion reads from 536 normal donors and 1,873 patients encompassing 21 cancer types, identified ~300 million such uncharacterized reads, and using a distinctive approach *de novo* assembled 2,550 novel human transcripts, which mainly represent long noncoding RNAs. Of these, 230 exhibited relatively specific expression or non-expression in certain cancer types, making them potential markers for those cancers, whereas 183 exhibited tissue specificity. Moreover, we used lentiviral-mediated expression of three selected transcripts that had higher expression in normal than in cancer patients and found that each inhibited the growth of HepG2 cells. Our analysis provides a comprehensive and unbiased resource of unmapped human transcripts and reveals their associations with specific cancers, providing potentially important new genes for therapeutic targeting.

**Keywords** cancer-associated transcripts; long noncoding RNAs; unmapped sequencing reads

**Subject Categories** Chromatin, Epigenetics, Genomics & Functional Genomics; Cancer

**Mol Syst Biol. (2015) 11: 826**

## Introduction

Central repositories of cancer genome data comprise enormous amounts of high-throughput sequencing data, enabling researchers to identify clinically relevant genes and genetic variants with prognostic, diagnostic, and therapeutic potential (Collins & Barker, 2007; International Cancer Genome Consortium *et al*, 2015). Sequencing data across multiple cancer types have previously been analyzed to determine expression profiles, epigenetic marks, and the sequence variations that occur in cancers (van't Veer *et al*, 2002; Greenman *et al*, 2007; Heintzman *et al*, 2007; Cancer Genome Atlas Research Network, 2008; Sotiriou & Pusztai, 2009; Stratton *et al*, 2009; Cancer Genome Atlas Research Network *et al*, 2013). However, a small albeit significant portion of sequencing data has remained "un-annotated" due to its unmappability to the human reference genome. In fact, crucial classes of RNAs such as microRNAs and long noncoding RNAs were only annotated over the past few years, and this motivated us to comprehensively analyze and annotate a large compendium of uncharacterized RNA-sequencing (RNA-Seq) reads in order to identify "new" transcripts. The reads we analyzed comprised multiple cancer types as well as matching normal tissues. Here, we provide the first *de novo* assembly of such uncharacterized reads, thereby creating a resource of thousands of previously missed human transcripts. The sequences of these transcripts were not available (i.e., not assembled) in the human reference genome, but we could map them to the chimpanzee/gorilla genomes, and then, using human homologs of their adjacent genes, we could predict their relative locations in the human genome. Moreover, we associated some of the transcripts with specific cancers and in several cases with the expression of adjacent genes. Additionally, we found histone H3K4me3 and H3K27ac transcription and regulatory-associated marks at genomic loci for > 150 of the transcripts, which suggests they are enhancer RNAs. Finally, we provide initial data of the functional significance for three newly assembled transcripts.

## Results

### Characterizing unmapped sequences and discovery of novel transcripts

To characterized unmapped sequencing reads, we developed a new data processing pipeline (Fig 1A). Briefly, we obtained RNA-Seq data from 1,873 patients encompassing 21 cancer types from The Cancer Genome Atlas (TCGA) data portal. To assure data quality and consistency, we only used data from paired-end Illumina platform libraries generated from solid primary tumors after May 2012 (Fig 1B; cancer abbreviations are in Table EV1A and information from TCGA for cancer samples is in Table EV1B). In addition to

Laboratory of Molecular Immunology and the Immunology Center, National Heart, Lung, and Blood Institute, National Institutes of Health, Bethesda, MD, USA
*Corresponding author. Tel: +1 301 496 0098; E-mail: wjl@helix.nih.gov

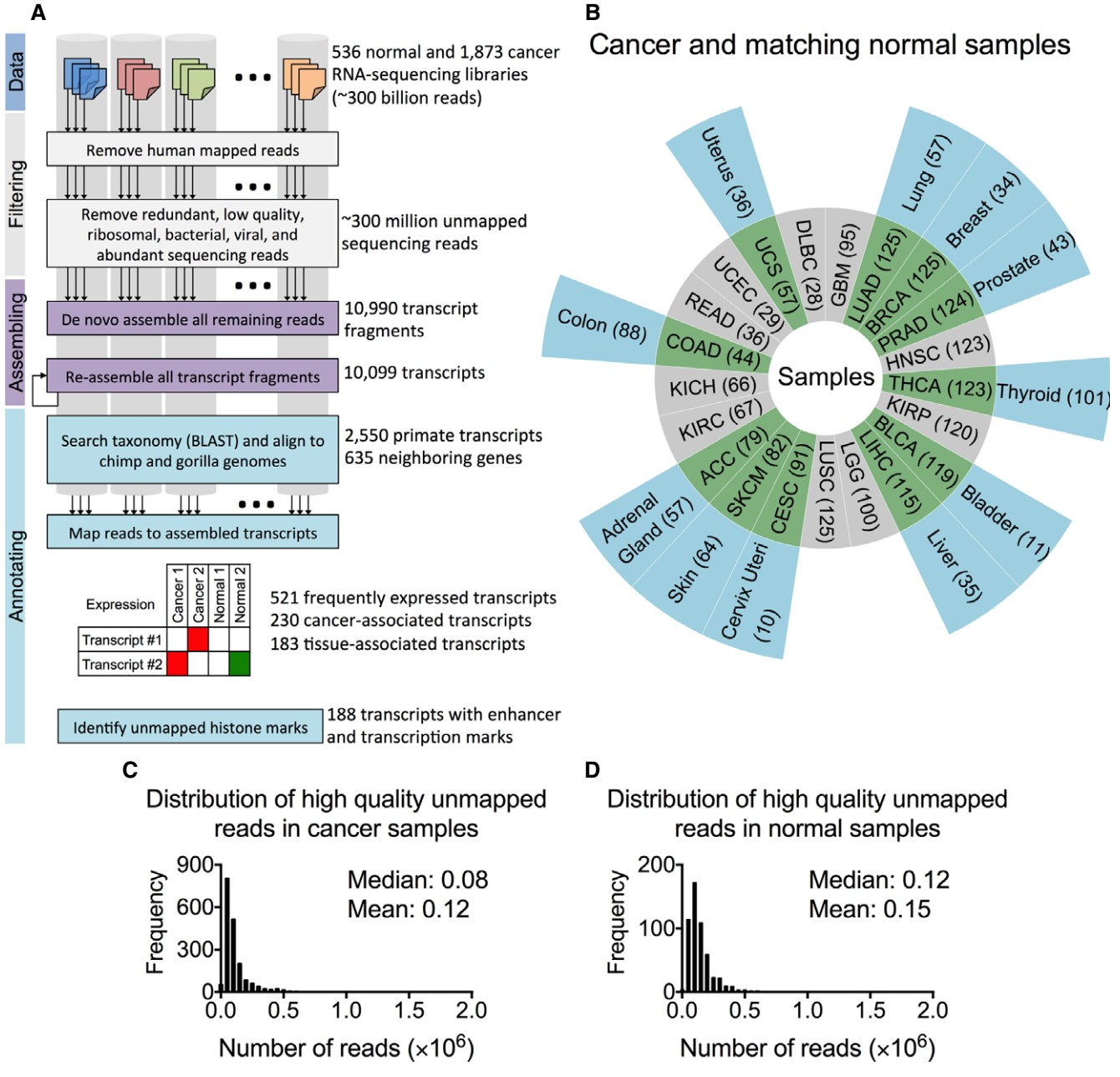

**Figure 1. Characterizing unmapped sequences.**

A Data processing strategy for identifying missed transcripts. Sequencing reads from cancer and normal samples were mapped to the human genome and transcriptome. Abundant reads (e.g., polyA, polyC, ribosomal RNAs, phage) and low-quality reads were discarded, and reads that mapped to known viral and bacterial sequences were removed. The remaining unmapped reads were pooled and *de novo* assembled to obtain previously missed transcripts. The newly assembled transcripts were annotated by their over- or under-representation in each cancer and the presence of histone marks in their genomic loci. For the illustrated 2 transcripts, one was expressed only in cancer and one in both cancer and normal tissues.

B Cancer types (inner donut) and matching normal tissue (outer donut). The numbers of samples are in parentheses. The abbreviations for the different cancer types are in Table EV1A.

C, D Distribution of high-quality unmapped sequencing reads across all cancer (C) and normal (D) samples after screening as described in (A).

these datasets, we obtained RNA-Seq data from 11 corresponding normal tissues/organ types from 536 healthy donors from the Genotype-Tissue Expression (GTEx) database (GTEx Consortium, 2013) (Fig 1B; see Table EV1C and D for information on these samples). Together, these libraries comprised over 300 billion sequencing reads. To focus on unmapped reads, we removed all reads that aligned to the human reference genome (hg19, GRCh37) and transcriptome (see Materials and Methods). We also eliminated from further analysis low-quality reads, duplicate reads, and reads that mapped to cloning vectors, abundant sequences, haplotypic regions

 

---

**Figure 2.** *De novo* **assembly of previously missed transcripts and characterization.**

A    Length distribution of 10,099 assembled transcripts.

B    Relative abundance of repetitive elements in the entire human genome versus genomic loci of newly assembled human transcripts. Shown is the fraction of the entire human genome (black bars) and newly assembled human transcript sequences (gray bars) corresponding to each indicated repeat family. Human genome repeats were downloaded from RMLib 20120124 from RepeatMasker (Smit *et al*, 1996–2010) genomics datasets. The primate transcripts were screened by RepeatMasker open-4.0.5, RMLib: 20140131.

C    GC content distribution of assembled transcripts. The bar graph shows the frequency at each GC percentage. The pie charts display the inferred taxonomy of transcripts with GC content ≤ 65% and > 65%. *P*-value (****$P$ < 0.0001) was calculated by chi-square test with 3 degrees of freedom.

D    Pairwise alignment dot plots. Each panel shows the pairwise alignment between the indicated sequences. Sequences were sorted based on their chromosome and position and then concatenated prior to the alignment. The red dots in the top-right panel correspond to the alignments absent in hg38 as well as hg19 (see Materials and Methods for details).

E, F    PCR amplification of randomly selected transcripts from the human genomic DNA (E) and mixed cDNA (F). Shown are 8 transcripts that were absent in both hg19 and hg38. All bands were validated by classical Sanger-based DNA sequencing.

G    Predicted coding potential of assembled transcripts. For 2,550 transcripts identified by aligning to nonhuman primates, the stacked bars show the fraction of transcripts with predicted level of coding potential, based on the use of two computational methods, CPAT (Wang *et al*, 2013) and CPC (Kong *et al*, 2007).

---

(Li *et al*, 2010) of the primary reference assembly, and known bacterial or viral sequences, only retaining those with both ends unmapped (< 1% per library in average; see Materials and Methods). The above screening resulted in a collection of ~300 million high-quality unmapped reads remained from cancer (Fig 1C) and normal samples (Fig 1D).

To determine which of the unmapped reads overlapped and could be assembled to form longer transcripts, we pooled them together and used ABySS (Simpson *et al*, 2009) and an in-house re-assembly program to *de novo* assemble 10,099 long transcripts (Fig 1A and Materials and Methods; see Table EV2 for the sequences of all assembled transcripts). These assembled transcripts had a median length of 404 bases (Fig 2A), and as compared to the entire human genome (Smit *et al*, 1996–2010), they were generally similarly or less enriched in repeats (Fig 2B). These transcripts had a bimodal GC content distribution (Fig 2C), and when "BLASTed (Altschul *et al*, 1990)," the transcripts with higher GC content (> 65%) were enriched in sequences with short nucleotide matches (21–23 bp) to known bacterial sequences but had insufficient identity to be eliminated in the initial bacterial screening; conceivably, some of these might represent new or divergent bacterial species. Conversely, the transcripts with GC content less than 65% were enriched in primate sequences (Fig 2C; Venn diagrams). In total, 2,550 (of 10,099) newly assembled transcripts had sequence homology to primates, with the majority (~81%) having > 90% nucleotide match to chimp and/or gorilla genomes based on BLAT (Kent, 2002) (Fig 1A and Materials and Methods). Given that the constituent reads were selected as unmapped, despite their presence in primates, these assembled transcripts did not align to the human reference genome assembly (Fig 2D) (see Table EV3 for detailed information on alignment scores to chimp, gorilla, and human genomes) or to recently characterized long noncoding RNAs (Iyer *et al*, 2015).

Because all transcripts were derived from TCGA and GTEx RNA-Seq data, we hypothesized that the chimp/gorilla matching transcripts were indeed human sequences but that they were missed from the human reference genome during sequence assembly. Indeed, during the course of the study, the genomic sequences corresponding to nearly half of these transcripts were correctly assembled and became available in hg38 reference genome (see Materials and Methods; Table EV3), but even in hg38 they were still not annotated as transcripts. Moreover, when we selected 8 transcripts with significant alignment to the chimp genome but no alignment to the human reference genome (Table EV4), we validated all

8 transcripts by PCR amplification using specific primers (Table EV5) followed by Sanger sequencing from genomic DNA (Fig 2E) as well as from a cDNA mixture derived from cancer and normal tissues (Fig 2F; see Materials and Methods for information on the cDNA mix). We next sought to estimate how many additional such transcripts in those tissues remain to be found under the same assembly conditions (e.g., length cutoff and sequence complexity). Whereas the precise number of missed transcripts is unclear, simulating the number that we would anticipate to be identified from a given number of RNA-Seq samples suggested that we have indeed found the majority of such transcripts (Fig EV1). Nearly 32% (806 of 2,550) of these transcripts were aligned to discontinuous blocks of target sequences or gaps, indicative of exons and introns in these transcripts (Appendix Fig S1; see also Table EV3, column X). Using CPAT (Wang *et al*, 2013) and CPC (Kong *et al*, 2007) computational methods, which predict the coding potential of a given transcript based on various sequence features including open reading frames and the alignment to known protein domain families, we found that > 95% of the transcripts were classified as long noncoding RNAs (Kapranov *et al*, 2007) (Fig 2G), a class of RNAs with broad functions (Mercer *et al*, 2009).

## Cancer- and tissue-associated transcripts

We next evaluated whether any of our newly discovered human transcripts are associated with specific cancers and/or tissue types. We mapped all the RNA-Seq libraries to the newly assembled transcript sequences to determine the expression level (Fig 3A and Table EV6A and B) and frequency (Table EV6C and D) in the 11 cancer types for which we had corresponding normal tissue samples. We identified 521 (of 2,550) transcripts that were expressed in more than 10% of tissue samples of the same type, and we herein call these frequently expressed transcripts (Fig 3B). These transcripts were expressed at lower levels and were shorter than protein-coding transcripts but similar in expression level and sequence length to that of known long noncoding RNAs (Fig EV2). Strikingly, 47% (243 of 521) of these had significantly different expression frequency (higher, lower, or mixed) in cancer versus corresponding normal tissue samples and thus were considered as cancer-associated transcripts (two-tailed Wilcoxon signed-rank test *P*-value < 0.0001). For example, transcript asm|33042290 (see Materials and Methods for nomenclature) was expressed in 49% of adrenal cortical cancer patients but only in 10% of normal adrenal

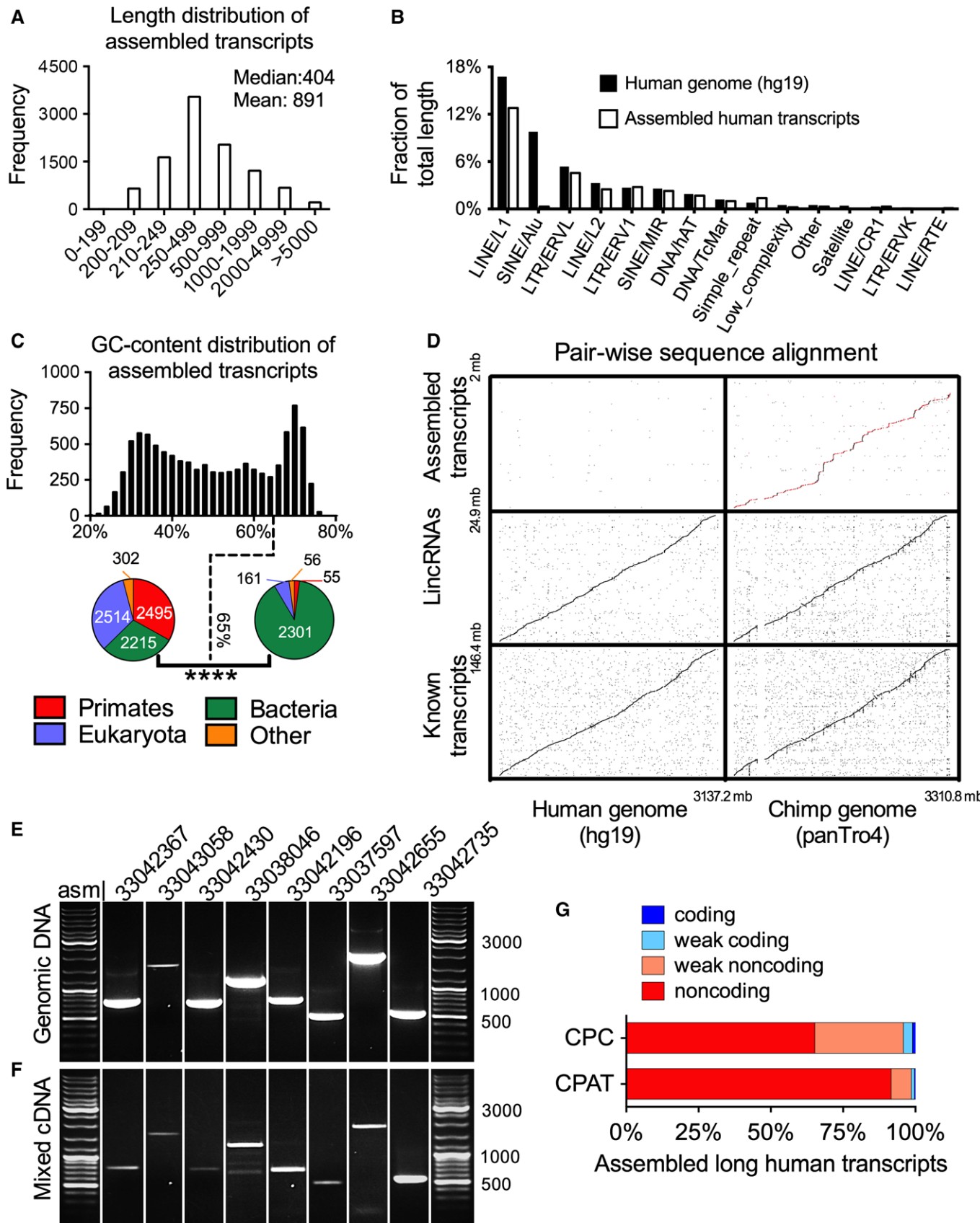

**Figure 2.**

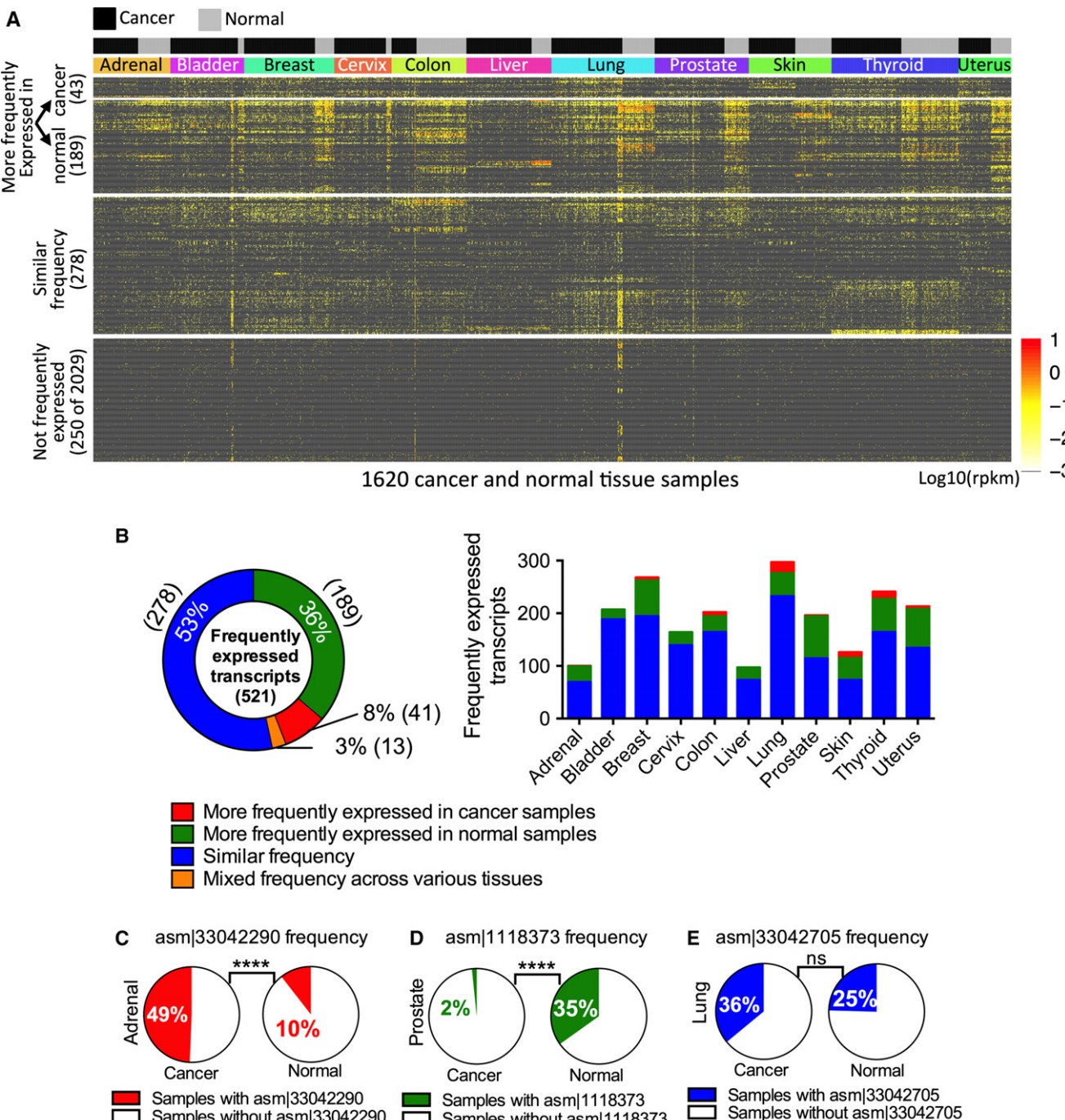

**Figure 3.  Cancer and tissue specificity of newly assembled transcripts across 11 tissues.**

A    The expression profiles of newly assembled human transcripts (rows) in various cancer and corresponding normal tissue samples (columns). Transcripts are grouped into four categories: those that are significantly more frequently expressed (two-tailed Wilcoxon signed-rank test, *P*-value < 0.0001) in cancer than normal, those that are significantly more frequently expressed in normal, those with similar frequency, and those that are not frequently expressed. The last category contains 250 (of 2,029) randomly selected representatives. Transcripts are defined as "frequently expressed" if they are expressed in more than 10% of samples of the same tissue.

B    The percentages of transcripts that are significantly more frequently expressed (two-tailed Wilcoxon signed-rank test, *P*-value < 0.0001) in normal or cancer tissue samples, have mixed expression frequency (higher in some cancers and higher in some other normal tissues) or have similar frequency of expression in cancer and normal samples. The number of transcripts in each category is shown in parentheses. The stacked bar chart on the right shows the number of frequently expressed transcripts in each category for 11 different tissues.

C–E   Three representative transcripts that are more frequently expressed in cancer than in normal tissues (C), more frequently expressed in normal tissues than in cancer (D), or have similar frequency of expression (E). For each transcript, the left (right) pie chart illustrates the frequency of samples in a specific cancer (normal) tissue. *P*-values (****P* < 0.0001) were calculated by two-sided Fisher's exact test.

gland samples (Fig 3C), and transcript asm|1118373 was expressed in 35% of normal prostate samples but only in 2% of prostate adenocarcinoma patients (Fig 3D). Interestingly, 83 cancer-associated transcripts had significant differential expression frequency in only one or two cancer types, thus making them relatively specific to certain cancer(s) (e.g., asm|33042290 was relatively specific for adrenal and skin cancers; see Appendix Fig S2). Conversely, 53% (278 of 521; Fig 3B) of frequently expressed transcripts had similar expression frequency in cancer and normal samples (e.g., asm|33042705 in Fig 3E), but most of these (183 of 278) were specific to a few tissue types and thus were considered as tissue-specific transcripts (see Table EV7A and B for complete results).

We next investigated whether there was an association between the expression of frequently expressed transcripts and their adjacent genes, as such "*cis*" associations might indicate a mechanism of action for some of these transcripts, analogous to reports for other long noncoding RNAs (Brockdorff *et al*, 1992; Nagano *et al*, 2008; Orom *et al*, 2010). To this end, we predicted the genomic neighborhood

and the nearest gene for 88% of the frequently expressed transcripts using synteny (Pruitt *et al*, 2014) between human and chimp and/or gorilla (see Materials and Methods); for the remaining transcripts, we could not locate such neighboring genes in chimp or gorilla genomes. We then determined whether the expression of the adjacent gene was significantly different in samples expressing a transcript than in samples not expressing the transcript, using both cancer and corresponding normal tissue samples. We observed 149 such strong differences (two-tailed Wilcoxon signed-rank test *P*-value < 0.0001) (Fig 4A; see Table EV7C for complete results). Interestingly, in 85 of these transcripts, elevated expression of the adjacent gene was observed in cancer but not normal samples, indicating cancer-specific elevation. For example, for samples expressing transcript asm|33042290, mRNA expression of *RAB38*, the predicted adjacent gene, was higher in cancer but in not normal samples (Fig 4B). In contrast, in 64 transcripts, the augmented expression of adjacent genes was observed in normal or both cancer and normal samples, indicating tissue-specific elevation. For example, mRNA

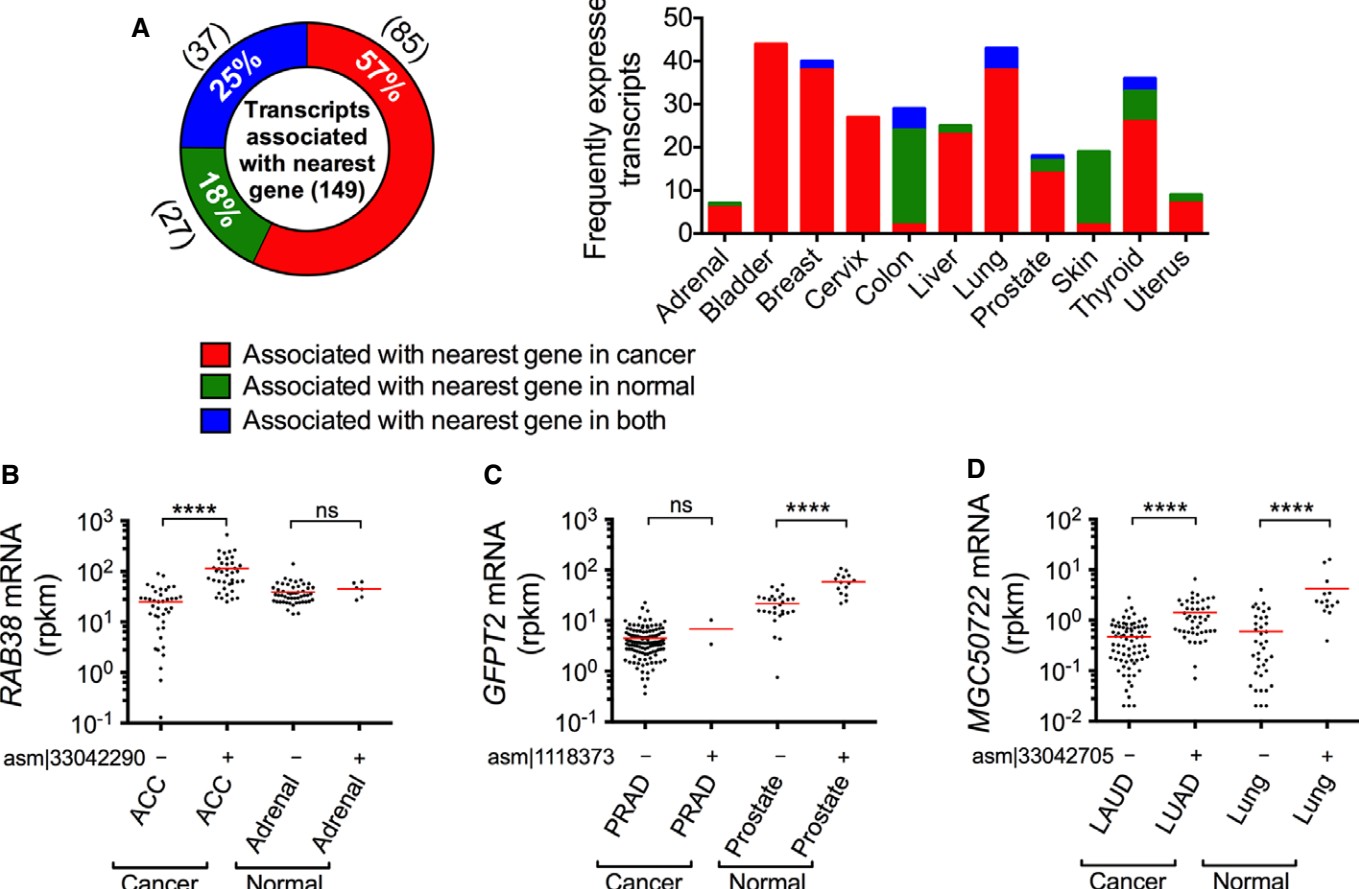

**Figure 4. Nearest gene association.**

A    Percentage of transcripts associated with the expression of the adjacent genes in cancer, normal tissue, or both types of samples. The stacked bar chart on the right shows the number of frequently expressed transcripts that are associated with their nearest gene in cancer, normal, or both types of samples.

B–D    Three representative transcripts that are associated with their nearest gene expression in cancer (C), normal (D), or both (E) cancer and normal samples. The transcripts are chosen to be the same as in Fig 4C–E. Significance of the association of the expression of the gene adjacent to the indicated transcript in samples expressing (+) versus those not expressing (−) the transcript; each panel shows these separately for cancer and normal samples. *P*-values (****P < 0.0001) were calculated by two-sided Wilcoxon signed-rank test. The red lines show the median value.

    

expression of *GFPT2* was higher in samples expressing transcript asm|1118373 only in normal samples (Fig 4C), whereas *MGC50722* was higher in samples expressing transcript asm|33042705 in both cancer and normal samples (Fig 4D). In several cases, both high differential expression frequency (Fig 3C–E) and adjacent gene association were observed (Fig 4B–D).

### Active transcription and regulatory marks in the genomic loci of assembled transcripts

To assess the transcriptional and regulatory activity of our novel transcripts, we examined histone H3K4me3 and H3K27ac modifications, which are typically associated with active transcription and enhancer activity (Bernstein *et al*, 2005; Heintzman *et al*, 2007), respectively. These histone mark data were not available for the cancer (TCGA) or normal (GTEx) samples we analyzed, so we downloaded histone H3K4me3 and H3K27ac ChIP-Seq data from 11 cancer cell lines, corresponding to breast, cervix, colon, liver, lung, and prostate tissues (Ernst *et al*, 2011; Akhtar-Zaidi *et al*, 2012; Lin *et al*, 2012; Tropberger *et al*, 2013; Hazelett *et al*, 2014; Ho *et al*, 2014; Rhie *et al*, 2014). Moreover, due to the absence of our transcript sequences in the genome assembly, traditional methods of peak (histone modification) calling such as MACS (Zhang *et al*, 2008) were not applicable. Thus, to detect significant levels of histone marks, we developed a statistical method (see Materials and Methods) that could strongly segregate traditional peaks from non-peaks (Appendix Fig S3). The presence of H3K4me3 and H3K27ac marks provides additional evidence of transcriptional and regulatory activity of these transcripts.

We detected significant levels (Poisson statistics *P*-value < 0.0001; see Table EV8) of H3K4me3 and/or H3K27ac marks in the genomic loci of 188 transcripts in one or more cancer cell lines (Fig 5A). H3K27ac was identified for 69% of these transcripts, H3K4me3 was identified for 12%, and both marks were found at 19% of the genomic loci for these transcripts (Fig 5A). The presence of enhancer marks at the genomic loci of these transcripts is consistent with their being enhancer RNAs (Kim *et al*, 2010), a class of noncoding RNAs with potential *cis*- or *trans*-regulatory role(s). Approximately one-third (61 of 188) of transcripts had significant histone marks across multiple tissues and cancer cell lines (Fig 5B; see Table EV8 for histone mark data on all transcripts). Interestingly, 42 transcripts had histone marks at their genomic loci in at least one cancer cell line that corresponded to the type of cell/tissue in which the transcript is frequently expressed (Fisher's exact test *P*-value < $3 \times 10^{-5}$; Table EV9; Fig 5B, dark gray highlighted transcripts in left-side bar). Most of these transcripts were differentially

expressed in cancer and normal samples, and several were associated with the expression of the adjacent gene (Table EV10). For example, asm|33039309, which has a significant H3K27ac mark at its genomic locus in the LNCap prostate cancer cell line (Fig 5C), was more frequently expressed in prostate cancer than in normal samples (54% vs. 31%; Fig 5D), with increased expression of its adjacent gene, *GADD45G*, in normal or cancer samples that express this transcript (Fig 5E). Moreover, some of the associated adjacent genes are implicated in cancer. For example, *PIM3* is a proto-oncogene that enhances pancreatic cancer growth by modulating tumor vasculogenesis (Kuang *et al*, 2013; Liu *et al*, 2014); *PTP4A3* can promote cancer metastasis particularly in colorectal cancers (Al-Aidaroos & Zeng, 2010; Cramer *et al*, 2014; Huang *et al*, 2014); and *TOMM20* expression is associated with tumor size in gastric cancer (Zhao *et al*, 2014).

### Transcripts asm|33038046, asm|33042430, and asm|33042735 regulate cell growth

To begin to elucidate the biological actions of our newly assembled transcripts, we selected three out of the 8 PCR-validated transcripts, each of which was more frequently expressed in normal than in cancer samples (Fig 6A, qPCR confirmation in Fig EV3). For example, asm|33042430 was present in 19% of all normal samples, whereas only 10% of cancer samples express this transcript (two-tailed Wilcoxon signed-rank test *P*-value < $2e^{-6}$). To determine whether these transcripts regulate cell growth, we used lentiviral-mediated transduction of HepG2 carcinoma cells (Fig EV4). In each case, constitutive expression of these transcripts significantly inhibited cell growth by 3 days (Fig 6B; left panel), with no change in cell death (Fig EV5). The behavior of non-transduced cells was similar to wild-type cells (Fig 6B; right panel). Interestingly, the cells expressing these RNAs also exhibited an increase in cell size, as assessed by flow cytometry (Fig 6C), suggesting cell division cycle interruption by these transcripts. Our results indicate the biological significance of at least a subset of these newly assembled transcripts.

## Discussion

We have identified more than 2 million bases of novel human transcripts that are absent in the reference human genome, underscoring the importance of deep mining of sequencing data to comprehensively characterize unmapped reads. Some of these transcripts, in particular single exonic ones, might originate from promoter upstream transcripts (PROMPTs), or enhancer regions,

---

**Figure 5. Histone marks at genomic loci of newly assembled transcripts.**

A   Distribution of H3K4me3 and H3K27ac histone marks in genomic loci of 188 transcripts with significant histone marks in one or more cancer cell line(s). Numbers in parentheses indicate the number of transcripts in each category. The stacked bar chart on the right shows histone mark distributions in each tissue type.
B   Heat map of histone mark profiles for 188 transcripts across multiple tissues. For each transcript (rows), H3K27ac and H3K4me3 marks are shown in the indicated tissues (columns). The intensities of the colors are proportional to the negative log *P*-value of significance in each row. The gray-scale left-side bar highlights significant histone marks on transcripts that are frequently expressed in a tissue that matches that of the cancer cell line. When histone marks from multiple cancer cell lines existed for a tissue, the most significant one is shown for that tissue (see Table EV8 for complete results).
C   H3K27ac and H3K4me3 marks on the genomic locus of the transcript asm|33039309 in multiple tissues (when multiple cancer cell lines exist for a tissue, histone marks in a representative cancer cell line are shown).
D   The expression frequency of asm|33039309 in cancer and normal prostate samples. *P*-value (**\**P* = 0.008) was calculated by two-sided Fisher's exact test.
E   mRNA expression of the neighboring gene *GADD45G* for samples with or without asm|33039309 expression in cancer and normal prostate samples. *P*-values (**\***P* = 0.001, *\**P* = 0.02) were calculated by two-sided Wilcoxon signed-rank test. The red lines show the median value.

---

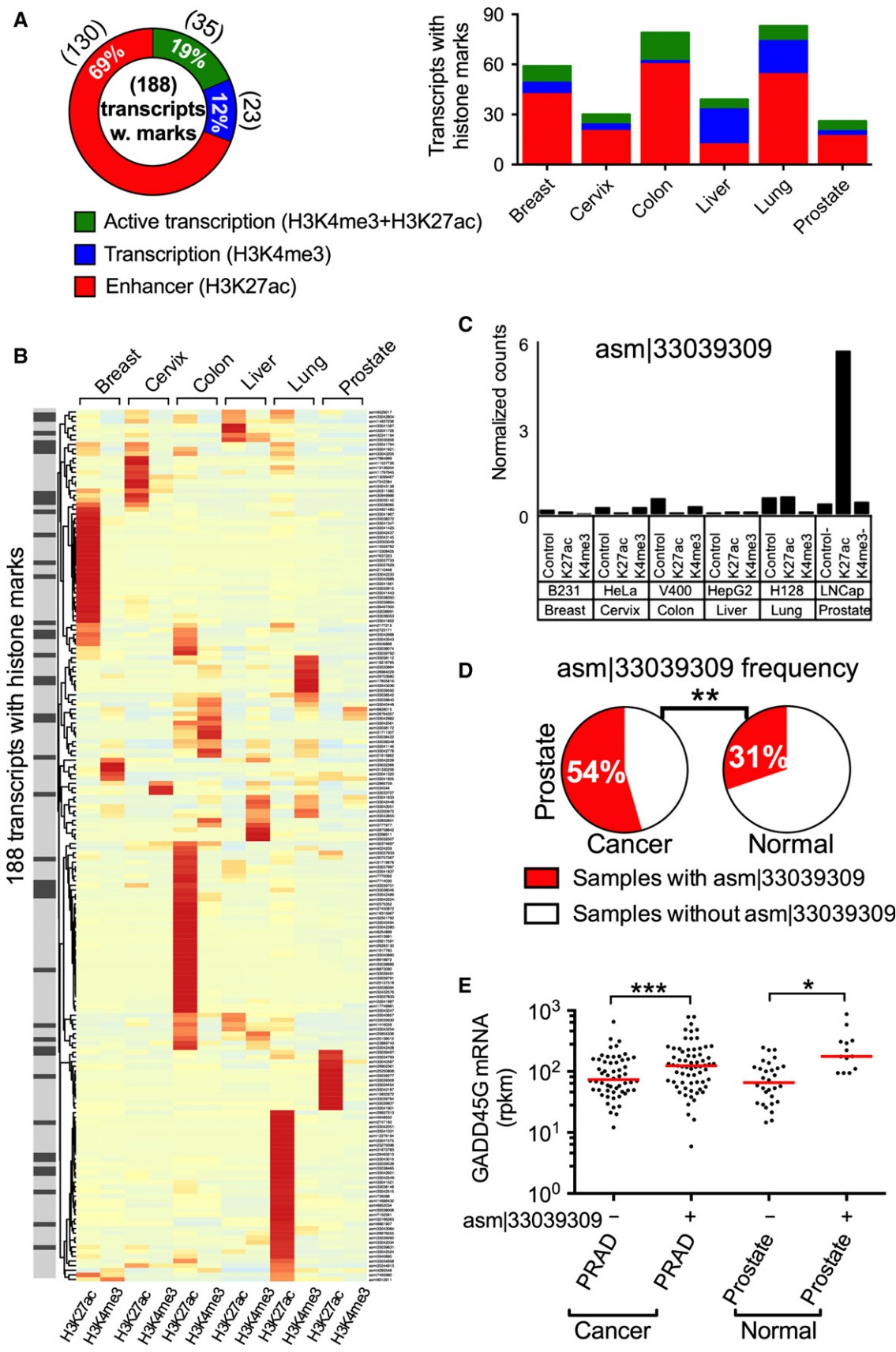

**Figure 5.**

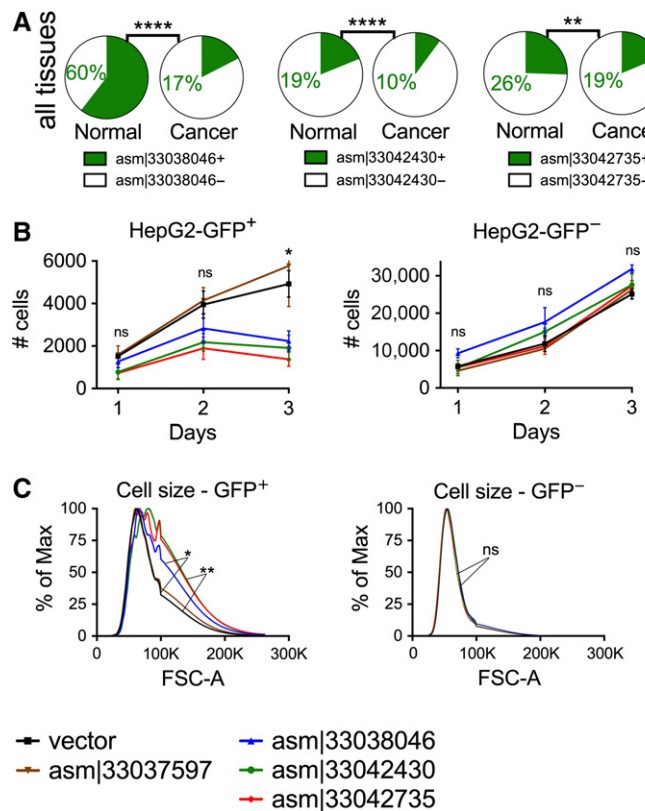

**Figure 6. Transcripts asm|33038046, asm|33042430, and asm|33042735 regulate cell growth in HepG2 cells.**

A  For each of these transcripts, two pie charts are shown with the frequency of expression in normal (left pie chart) or cancer (right pie chart) samples. Shown are the percentage of samples expressing (green) or not expressing (white) the transcript. asm|33038046, asm|33042430, and asm|33042735 are each more frequently expressed in normal than cancer samples. *P*-values (****$P < 0.0001$, ***$P = 0.002$) were calculated by two-sided Fisher's exact test.

B  Lentiviral-mediated overexpression of transcripts in HepG2 carcinoma cells. Shown are the cell numbers over the course of 3 days after cellular transduction with vector or transcripts. Control transcript asm|33037597 did not affect cell growth. The vector expresses GFP; thus, GFP⁺ cells represent the cells successfully transduced. GFP⁺ and GFP⁻ cells are shown in the left and right panel, respectively. *P*-values (*$P = 0.02$) were calculated by two-sided *t*-test. Error bars show standard error of the mean.

C  Cell size on day 3 after transduction of transcripts or empty vector in HepG2 carcinoma cells. Shown is the cell size, as measured by flow cytometry. Data represent the average of two replicates. *P*-values (*$P = 0.011$, **$P = 0.007$) were calculated by two-sided Wilcoxon signed-rank test.

Data information: (B, C) Data shown are representative of three independent biological experiments.

and although they might lack immediate function (Andersson *et al*, 2014), the uncovered corresponding genomic sequences could harbor regulatory elements controlling gene expression. Aligning the assembled transcripts to the chimp and gorilla genomes enabled us to determine their relative genic neighborhoods in the human genome and in many cases to associate these conserved transcripts with expression of the neighboring gene. Our use of cross-species homology is an approach broadly applicable to the assembly and annotation of missed regions in the genomes of a range of species.

Although the functions of the transcripts we have discovered remain to be elucidated, the cancer/tissue association, correlation

with neighboring genes, and the presence of histone H3K4me3 and H3K27ac marks are tantalizing, and our lentiviral expression studies of three selected transcripts indicate roles in cell growth regulation, consistent with their higher expression frequency in normal than cancer patient samples. Collectively, these results provide compelling evidence to support further studies of the roles of these novel transcripts in normal physiology, in the development and progression of cancer, and potentially in other pathological situations as well.

## Materials and Methods

### Data processing pipeline

All cancer RNA-Seq BAM files (Binary sequence Alignment Map) were downloaded using GeneTorrent (https://cghub.ucsc.edu/) from dbGaP study accession phs000178. The normal RNA-Seq data were downloaded using SRA Toolkit from dbGaP study accession phs000424. All data were handled in accordance with the Data Use Certification Agreement. The unmapped reads from BAM files were initially extracted using samtools (Li *et al*, 2009). The low-quality reads were removed using Stacks:process_shortreads program (Cole *et al*, 2005), with parameters were set to [-c -q -s 17 -w 0.15 –filter_illumina –no_read_trimming]. The potential PCR clones in each library were removed using Stacks:clone_filter program (Cole *et al*, 2005) to keep unique reads. The reads in each library were further screened for mapping to reference genome (hg19, GRCh37 including all alternative haplotypes), transcriptome (GENECODEv19 comprehensive transcripts (Harrow *et al*, 2012); RefSeq genes; human all mRNAs (Pruitt *et al*, 2005); USCS genes (Hsu *et al*, 2006); Ensemble genes (Hubbard *et al*, 2002); lincRNAs (Trapnell *et al*, 2010); and human ribosomal RNA sequences); abundant sequences (vector sequences [http://www.ncbi.nlm.nih.gov/tools/vecscreen/univec/]; phage sequences (Leinonen *et al*, 2011); and polyA/C sequences), bacterial rRNA sequences (Cole *et al*, 2005), and bacterial and viral genomic sequences (Leinonen *et al*, 2011). The data for GeneCode, RefSeq, UCSC, and Ensemble transcripts were obtained using the Table Browser from UCSC on March 2014. The mapping was performed using the Burrows–Wheeler Aligner (BWA (Li & Durbin, 2009)) with the following parameters [$l = 28$, $k = 3$, $n = 0.1$, $q = 20$]. It is notable that BWA is a DNA aligner and does not directly take splicing into consideration; therefore, we included transcriptomic sequences, as mentioned above, to recover the reads that may cover splicing junctions. All sequences and transcripts were combined using the nrdb (ftp://ftp.ncbi.nlm.nih.gov/pub/nrdb/) program to remove trivial redundancies before creating the BWA index database. In each step, only paired-end reads with both ends unmapped were kept.

### Pairwise alignment

All the sequence pairwise alignments were performed using the MUMmer3.23 (Kurtz *et al*, 2004) program with [-mum -b -c] parameters. The rest of the parameters were kept as default. The human (hg19 and hg38), chimp (panTro4), and gorilla (gorGor3) genomes were downloaded from UCSC genome browser. The "transcripts"

corresponded to 47,447 mRNA sequences present in RefSeq for hg19 genome assembly. The lincRNAs sequences (Trapnell *et al*, 2010) were obtained from UCSC genome browser. The sequences in each genome, transcripts from RefSeq, or lincRNAs were sorted and concatenated (with 50 bp N-gap) based on their chromosome and locations prior to the MUMmer run. The chromosome and the location of the transcripts were determined by their alignment to the chimp genome using BLAT. The RefSeq transcript and the lincRNA sequences were oriented 5′-to-3′ before concatenation.

### Expression data analysis

All mRNA expression levels for coding genes in cancer and normal samples were re-calculated by RSEMv1.2.14 (Li & Dewey, 2011)/ Bowtie1 (Langmead *et al*, 2009) pipeline to remove any systematic bias that stems from the difference between the pipelines used for estimating the gene expression in TCGA and GTEx. The RSEM parameters were set to [–bowtie-n 1 –bowtie-m 100 –seed-length 28]. The expression level for newly assembled transcripts is calculated by RPKM values using the equation RPKM = $(10^9 \times C)/(N \times L)$, where C is the number of unique high-quality reads that mapped to the transcript, N is the total number of unique high-quality reads in the library, and L is the length of the transcript. Three bladder, 1 colon, and 8 lung cancer samples exhibit high-level expression for many transcripts (vertical bars in Fig 3A). This could be an artifact of batch processing of these samples (3 bladder, 1 colon, and 4 of the lung samples belong to sequencing plate A277, and the other 4 lung samples belong to the plate A278). However, given that these are only a small fraction of the total samples and retaining vs. omitting them has no significant impact on our results, we decided to keep them.

### Assembly and re-assembly

We pooled all the unmapped reads (> 516 million reads) together and ran the "ABySS-pe" program (Simpson *et al*, 2009) with default parameters. The nomenclature for newly assembled transcripts is generated by ABySS as "asm|", which stands for assembly, followed by a random number. Although it is possible that some of the reads were from low-abundance RNAs and thus difficult to distinguish from background, we minimized this potential problem by pooling unmapped reads from all cancer and normal samples to increase the read number and facilitate assembly of the transcripts. From the initial transcripts (scaffolds) generated by ABySS, those with length > 200 bp were classified as long transcripts. We chose this cutoff length as it is a standard length used for defining long noncoding RNAs, but we also observed ~5,000 shorter (100–200 bases) transcripts with a significant match to nonhuman primates, which are also absent in human reference genome and lack transcript annotation (data not shown). To ensure that initially assembled transcripts were "complete" and could not be further co-assembled, we developed an in-house program that joins a set of transcripts if there are supporting reads connecting them together. Briefly, we mapped all the unmapped reads to the long transcripts. For every pair of long transcripts, if there were more than five reads with one end mapping to one transcript and the other end mapping to the other transcript, the transcripts were joined. The orientation of the joins was obtained from the sequence arrangement in the paired-end

libraries manifested in the SAM file bitwise FLAG. Joined transcripts were examined for the possibility of higher order joining by iterating this process (e.g., pairs "a:b" and "b:c" would be joined to constitute "a:b:c"). Of 2,550 novel human transcripts, 221 were transcripts obtained from joining two or more transcript fragments, of which 176 were assembled in hg38 genome. Considering the hg38 assembly as the "gold standard", 165 transcripts (94%) were correctly reassembled, that is, the alignment of the transcripts to hg38 contained contiguous blocks of high-identity alignments. However, it is still possible that some of the newly identified transcripts may not represent the full-length cDNA and/or some of these transcripts may be a combination of overlapping and/or alternative splicing of a gene.

### BLASTing against the nt database

We utilized the "blastp" program of the locally installed blastall v2.2.26 and the nt.00-26 database to perform all the BLAST operations. We invoked BLAST with default parameters. Nearly 85% of the transcripts had significant BLAST match (E-value ≤ 0.05), whereas ~10% (1,100 of 10,099) of the transcripts had no significant BLAST match (E-value > 0.1) against nt database (Table EV3). The taxonomy information of the BLAST results (i.e., primate, bacteria, eukaryote, others) was obtained from NCBI taxonomy browser.

### Calling histone marks at genomic loci of newly assembled transcripts

For each transcript and each cell line, we calculated the significance of the change between the indicated histone mark and its corresponding input control using a Poisson statistics, 1-$F_{Poisson}(x; lambda)$, with *lambda* being the RPKM value of the transcript in the control library and *x* being the RPKM of the transcript in the corresponding histone H3K4me3 or H3K27ac library. RPKM values were obtained using the equation RPKM = $(10^9 \times C)/(N \times L)$, where C is the number of unique high-quality reads that mapped to the transcript, N is the total unique high-quality reads in the library, and L is the effective length of the transcript, defined as the length of a sub-sequence of the transcript that is covered by at least one read in any of the histone mark libraries. The significance cutoff was set at *P*-value ≤ 0.0001.

### Genic neighborhood determination using homology between human and chimp/gorilla genomes

We used the BLAT program to align the assembled transcripts to chimp (panTro4) and gorilla (gorGor3) genomes. All BLAT parameters were kept at default except minScore was set to 100. For each transcript, the best BLAT score was kept. We then used "bedtools closest" program to obtain the nearest gene in chimp/gorilla for each aligned transcript. Finally, the human homolog of the nearest gene was obtained using "Other RefSeq Genes" track from UCSC genome browser. It is notable that the relative location of some genes in human might be different from that of chimp/gorilla, which could introduce noise to the prediction of lncRNAs' nearest genes in human based on the alignment to the chimp/gorilla. To potentially quantify such a noise (i.e., how much the relative locations of genes

in human are similar/different to that of chimp/gorilla), we obtained all the genes (~19,000) in chimp/gorilla with known homology in human from http://www.ncbi.nlm.nih.gov/homologene. For each gene in human, chimp, and gorilla, we then found its nearest neighboring gene using "bedtools closest" program. We then checked what percentage of the nearest neighboring genes in chimp/gorilla are the same with the homologous genes and their nearest neighbors in human. We observed that 74.9% and 68.1% of relative gene neighborhoods are conserved between human and chimp or gorilla, respectively. Some transcripts had the same adjacent genes so that there were on average 2.8 transcripts for each predicted adjacent gene. A number of genes including *PECAM1, ERMARD, DHRSX, MGC50722, CD24, RAB7B, TCP10*, and *CHST15* each had more than 10 transcripts in their genomic neighborhood (Table EV7A), suggesting that the genomic neighborhoods of these genes are relatively poorly assembled and annotated in the available human reference genome. Although the human reference genome overall is more completely annotated than those from chimp and gorilla, the human transcripts we have identified in this study appear to correspond to regions that are better assembled in the chimp or gorilla genome assemblies.

Alignment to hg38. We aligned the transcripts to hg38 genome using BLAT. Of 2,550 newly identified transcripts, 1,473 (57%) have an assembly in hg38 and 1,260 (49%) have assembly with > 95% alignment (i.e., "correct assembly") between hg38 and the transcripts.

### HepG2 cell proliferation assay

The transcripts were amplified by PCR from HEK293T cells and cloned into lentiviral expression vector (pLVX-EF1alpha-IRES-ZsGreen1 Vector, Clontech). Transcripts expressing lentiviral particles were produced in HEK293T cells following the standard protocol. HepG2 cells were cultured in EMEM (ATCC), supplemented with 10% fetal bovine serum, in cell incubator at 37°C with 5% CO2. For lentiviral infection, HepG2 cells were cultured in 6-well plates for one day, followed by adding transcripts expressing lentivirus to the cells with 8 μg/ml of polybrene (Sigma). Twenty-four hours post-transduction, lentivirus-containing medium was removed and cells were recovered in normal growth medium for 48 h. Transduced HepG2 cells were subcultured into 24-well plates and allowed to attach for 16 h. Then, cells were washed and cultured in EMEM with 0.1% FBS for 24 h. On day 0, cells were labeled with CellTrace™ Violet (Life Technologies) according to the manufacture's recommendation and then incubated in full growth medium for 3 days. Each day, cells were taken off the plates by trypsin, washed, and stained with LIVE/DEAD® Fixable Near-IR Dead Cell dye (Life Technologies). Afterward, cell-counting beads (Life Technologies) were added to each sample and analyzed by FACS. Cell number in each sample was calculated and normalized by cell-counting beads number. Cell growth curve was plotted in Prism (GraphPad) software.

### cDNA samples

The cDNA samples used for validation in Fig 2F were obtained from AMSBIO company. They were all quality-controlled and DNase I pre-treated. The catalog (lot) numbers of samples are as follows:

C1234201-10 (B506159), C1234188 (B705038), C1234142 (B707257), C1234201 (B706051), C1235188 (A802038), C1235142 (A602226), C1234265 (B509030), C1235265 (A304109), C1234152 (B509033), C1234149 (B707258), C1234004 (B312074), C1235149 (A306002), C1235152 (A307028), C1235004-10 (A409321). The mixed cDNA is obtained by mixing cDNAs from seven tumor and matching normal tissues from adrenal gland, pancreas, liver, lung, kidney, thyroid, and prostate.

**Expanded View** for this article is available online:
http://msb.embopress.org

### Acknowledgements

The results shown here are in part based upon data generated by the TCGA Research Network (http://cancergenome.nih.gov/) and Genotype-Tissue Expression (GTEx) program. We also thank Genome Reference and Human Genome Sequencing consortiums for their contribution to human genome sequencing and assembly. We thank Drs. Claudia Kemper and Peng Li for valuable discussions and critical comments. This work was supported by the Division of Intra-mural Research, National Heart, Lung, and Blood Institute, National Institutes of Health and grant to MK from NIH (K22- KHL125593A). This study utilized the high-performance computing facility, helix/biowulf (http://biowulf.nih.gov/), at the National Institutes of Health.

### Author contributions

MK, MR, and WJL designed the study. RS, J-XL, and WL supplied material and expertise; MK performed all analysis; MR and MK undertook experiments; MK and WJL drafted and wrote the manuscript. All authors contributed to and approved the final manuscript.

### Conflict of interest

The authors declare that they have no conflict of interest.

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
