## [Review Process File · Molecular Systems Biology]

Comprehensive assembly of novel transcripts from unmapped human RNA-Seq data and their association with cancer

Majid Kazemian, Min Ren, Jian-Xin Lin, Wei Liao, Rosanne Spolski, Warren J. Leonard

Corresponding author: Warren J. Leonard, National Institutes of Health

Review timeline:

Submission date:	17 March 2015
Editorial Decision:	30 April 2015
Revision received:	05 June 2015
Editorial Decision:	10 July 2015
Revision received:	13 July 2015
Accepted:	14 July 2015

Editor: Maria Polychronidou

Transaction Report:

1st Editorial Decision

30 April 2015

Thank you again for submitting your work to Molecular Systems Biology. We have now heard back from the three referees who agreed to evaluate your manuscript. As you will see from the reports below, the referees acknowledge that the presented findings seem potentially interesting. However, they raise a series of concerns, which should be carefully addressed in a revision of the manuscript.

Without repeating all the points listed below, some of the more fundamental issues are the following:

- Further experimental analyses and additional controls are required in order to better characterize the biological function of the newly assembled transcripts and their differential expression in cancer vs normal tissues.
 - The methodology needs to be described in better detail so that it can be properly evaluated by the referees.
 - Additional analyses and/or clarifications are required, to provide further support for the conclusions deriving from the computational analyses.
-

Reviewer #1:

In this manuscript Kazemian and colleagues characterize unmappable sequencing reads derived from cancer genome atlas. The authors use a very clever and thorough approach to derive about 300 million high-quality unmapped reads from cancer and normal samples. These reads could be assembled into about 10000 transcripts of which about 2500 transcripts had sequence similarity to

other primate genomes. Interestingly 230 RNAs showed specific expression or non-expression in different cancer types. Furthermore the authors characterize the biological function of three of the newly assembled transcripts.

This manuscript is well written and the computational aspects of the work is appropriately conducted. The results are of general interest and timely. The analysis provides a comprehensive and unbiased resource of unmapped human transcripts.

Despite the exiting findings the reviewer has major concerns with the experimental characterization of the biological function of newly assembled transcripts. The analysis of these transcripts is superficial and poorly controlled and not appropriate for a journal like Molecular Systems Biology. In this study the authors study three newly assembled transcripts which are more frequently expressed in normal tissue than in cancers. Kazemian and colleagues use lentiviral transduction for the expression of the new transcripts and observe that constitutive expression of these transcripts significantly inhibited cell growth by three days compared to an empty vector only. Is the observed reduction in cell growth due to reduced proliferation or increased cell death? This question has not been addressed. It is the reviewer's opinion that the empty lentiviral construct is not an appropriate control for this experiment. The authors should have used a construct expressing an unrelated transcript of roughly the same length (e.g. the reverse sequence or the reverse complement of the identified transcripts) to compare whether the expression of a specific sequence or any expressed transcript, in general, causes the observed effect. In addition the authors should measure the expression level of the transcripts generated from lentiviral construct in comparison to a house keeping gene to inform the readers to which extent the transcripts are over-expressed. This is an important information since it seems that the three transcripts have low expression levels based on the reported RPKM values. Based on these findings it seems far-fetched to imply that the three characterized transcripts might have tumor suppressor activity as suggested in the abstract. Furthermore the authors should provide additional experimental support that the characterized transcripts are differentially expressed in cancer and normal tissue.

Minor points

- On page 5 the author make the statement that some of the newly identified transcripts were correctly assembled in the hg38 reference genome. It would be helpful to provide a more detailed analysis what fraction /covered bps of the newly assembled transcripts are found in the hg38 assembly
- Legend of Figure 6B should describe the GFP expressing lentiviral construct rather than legend for 6C.

Reviewer #2:

This is a potentially interesting study that aims at identifying new lincRNA genes by checking whether RNAseq panels contain reads that cannot be mapped to the human genome, and may have been missing from the assembly and thus annotation. To this end, the authors use a combination of alignments to the chimp and orang utan genomes and de novo transcript assembly. Topic-wise, this is therefore rather a genomics paper than a systems biology study.

The manuscript does not provide enough details about the approach and evaluation to be properly evaluated at its current stage.

Some major points:

- *) Did the authors check against human reads instead of the assembled reference genome?
- *) Is the complete transcript structure supported in chimp or just parts? Are these transcripts annotated in apes and/or detected in ape RNAseq data? How convinced are the authors that they were able to map the complete transcript -- and vice versa, that pooling so many libraries would not lead to the combination of distinct transcripts?
- *) all results need to be put in context with control set of known lincRNA or eRNA - expression, histone modifications, length etc.
- *) it is completely unclear how the authors mapped histone modifications -- what was the precise quantitative approach

to call a modification?

*) if many transcripts map to putative eRNA/PROMPT sites, then the authors effectively mapped promoters and enhancers

but not necessarily new genes. Most of these transcripts are evidence of transcriptionally active regions but whether they

have any function is highly debated. This should be clearly discussed, eg. in the context of studies such as (Andersson et al, Nature Comm 2014)

*) overexpression assays should be complemented by other assays -- these RNAs are largely lowly expressed and it is hard to gain insight from this approach alone.

minor points:

Fig 3A -- some expression samples appear suspicious, as they have high expression for many of the RNAs (visible as columns)

Reviewer #3:

The manuscript by Kazemian et al. identified novel non coding lncRNA in TCGA and GTEx samples. The authors used 1873 tumor samples and 536 normal tissue samples. In the analysis, the authors filtered all known transcript sequences across the species and then identified 10099 new transcripts. Based on GC, the majority of new assembled transcripts were removed as they had high GC content. 95% of the transcripts were assigned as lncRNAs. The authors also identified novel cancer associated transcripts and expression correlation cis associated gene expression. The authors have also tried to show the active transcription marks with novel lncRNA expression using cell line chip seq data.

Major Issues:

1. The authors have utilized BWA, which is a DNA aligner and does not take splicing into consideration. A splicing specific aligner should have been used for the alignment.
2. As the new human genome version hg38 is available, the authors should reanalyze the data using hg38 as reference genome.
3. The transcripts the authors claim are novel did not assemble to the genome and at least some of them could very well be variants of some other transcripts.
4. The authors have used an in-house re-assembly program for identification of new transcripts de-novo. The authors should provide more information on how the program works and also what is the false positive frequency.
5. The authors claim that new transcripts "were generally similarly or less enriched in repeats". This seems meaningless as it is difficult to assemble across repeats.
6. The authors claim to have found the majority of transcripts using some simulation. It is hard to know what is missing if the full content is not presented.
7. The authors have checked the neighborhood gene of lncRNA in gorillas and chimps and then checked if those genes are expressed in humans. It is unclear how fruitful this analysis will be as the location of genes in humans can be different than gorillas/chimps.
8. It seems strange that you could map reads to chimp/gorilla and determine the position in the human genome via homology, but not map the reads directly to the human genome. Could you just allow more mismatches to get more alignments to the human reference sequence?
9. To check the histone marks in lncRNA loci, the authors have used cell line chip seq data and aligned the data with newly identified transcripts. This is a clever idea but not sure if it is the correct method as many of the histone marks are not on the gene body and won't align with RNA seq.
10. In Figure 5 the authors should correlate the expression with histone marks.

11. The authors should compare the list of novel lncRNA with other recently published articles on identification of novel lncRNAs.

12. What is the sequence complexity of the unmapped reads? By eye, the sequences in Table S2 look OK (and the first sequence has a BLAST hit on a human BAC clone). However, it would be good to have some metric to indicate the degree to which the unmapped reads have more or less repetitive sequence than expected based upon the reference sequence. Figure S1 does address this issue, which alleviates concern that the de novo assembly mostly comes from low complexity sequences, but it is worth moving that to part of a main figure.

13. Less emphasis should be placed on the high GC content sequences that match bacterial sequences following the de novo assembly but not in the initial screen. For example, the authors should describe these as a filter for removed sequences rather than saying "Interestingly, we observed a bimodal GC-content distribution for these transcripts".

14. The author show some sequences align to hg38. What percentage of reads/transcripts?

15. The author state removed "reads mapped to cloning vectors". Is this simply checking for reads that match the adapter sequence? Given the paper specifically focuses on unmapped reads, the author should be extra careful about possible artifacts from library preparation.

1st Revision - authors' response

05 June 2015

We appreciate the reviewers' constructive criticism and valuable suggestions to improve the paper. We have performed additional experiments and have responded below to each the points that were raised.

Reviewer #1:

"The authors use a very clever and thorough approach to derive about 300 million high-quality unmapped reads from cancer and normal samples. ... This manuscript is well written and the computational aspects of the work is appropriately conducted. The results are of general interest and timely. The analysis provides a comprehensive and unbiased resource of unmapped human transcripts."

We are pleased with the reviewer's positive thoughts on the study.

Despite the exiting findings the reviewer has major concerns with the experimental characterization of the biological function of newly assembled transcripts. The analysis of these transcripts is superficial and poorly controlled and not appropriate for a journal like Molecular Systems Biology. In this study the authors study three newly assembled transcripts, which are more frequently expressed in normal tissue than in cancers. Kazemian and colleagues use lentiviral transduction for the expression of the new transcripts and observe that constitutive expression of these transcripts significantly inhibited cell growth by three days compared to an empty vector only. Is the observed reduction in cell growth due to reduced proliferation or increased cell death? This question has not been addressed.

As suggested by the reviewer, we have now used flow cytometry to assess cell viability in transduced (GFP⁺) vs. non-transduced (GFP⁻) cells over the course of three days. As shown in a new Supplementary Figure 8, we observed no significant change in viability of cells expressing vector vs. the indicated transcripts, suggesting that the observed reduction in cell growth is not due to increased cell death.

It is the reviewer's opinion that the empty lentiviral construct is not an appropriate control for this experiment. The authors should have used a construct expressing an unrelated transcript of roughly the same length (e.g. the reverse sequence or the reverse complement of the identified transcripts) to

compare whether the expression of a specific sequence or any expressed transcript, in general, causes the observed effect.

As suggested by the reviewer, we selected a transcript (asm|33037597) with similar size (~600 bp) to that of two of the tested transcripts, asm|33042430, and asm|33042735. As shown in Figure 6B, similar to the empty vector, transduction with asm|33037597 did not affect HepG2 cell growth. Moreover, when asm|33037597 used, the size of transduced (i.e., GFP⁺) cells is similar to the size of cells transduced with the empty vector (Figure 6C). These results suggest that the observed reduction in HepG2 cell growth is transcript-specific.

In addition the authors should measure the expression level of the transcripts generated from lentiviral construct in comparison to a house keeping gene to inform the readers to which extent the transcripts are over-expressed. This is an important information since it seems that the three transcripts have low expression levels based on the reported RPKM values.

As requested, we now show in Supplementary Figure 7 the expression level of the transcripts in the overexpression experiment shown in Figure 6. Using specific qPCR probes (Table S5) for each transcript and qPCR, we measured the relative expression (normalized to ACTB) in cells transduced with the indicated transcript.

Based on these findings it seems far-fetched to imply that the three characterized transcripts might have tumor suppressor activity as suggested in the abstract.

We agree with the reviewer that further experiments are required to establish tumor suppressor activity of these transcripts. Our main intention was to provide a proof of concept in terms of functionality of these previously missed transcripts and to show that there are still gaps in the current human genome assembly that may harbor functional sequences. We have modified the abstract and the manuscript on p. 10 by removing statements “, consistent with tumor suppressor activity” and “suggesting possible tumor suppressor activity”.

Furthermore the authors should provide additional experimental support that the characterized transcripts are differentially expressed in cancer and normal tissue.

We have performed expression analysis of the three transcripts used in Figure 6A by RT-qPCR on cDNAs from seven tumor and matching normal tissues from adrenal gland, pancreas, liver, lung, kidney, thyroid, and prostate. The cDNA samples were obtained from AMSBIO. Only one sample per tissue (cancer or normal) was commercially available. As shown in Supplementary Figure 6, all three transcripts have higher expression in normal than cancer samples, consistent with the observation that these transcripts are more frequently expressed in normal than cancer tissues (Figure 6A), except asm|33038046 had higher expression in thyroid cancer.

Minor points

- On page 5 the author make the statement that some of the newly identified transcripts were correctly assembled in the hg38 reference genome. It would be helpful to provide a more detailed analysis what fraction /covered bps of the newly assembled transcripts are found in the hg38 assembly.

As requested, we now provide more detailed information. Of 2550 newly identified transcripts, 1473 (57%) have an assembly in hg38, and 1260 (49%) have assembly with >95% alignment (i.e. “correct assembly”) between hg38 and the transcripts. We now clarify this in the Methods on p. 17. It is notable that even for those with correct genomic assembly in hg38, the transcript annotation is lacking.

- Legend of Figure 6B should describe the GFP expressing lentiviral construct rather than legend for 6C.

We have corrected this mistake and have now moved the description from 6C to 6B.

Reviewer #2:

“This is a potentially interesting study that aims at identifying new lincRNA genes by checking whether RNAseq panels contain reads that cannot be mapped to the human genome, and may have been missing from the assembly and thus annotation.” The manuscript does not provide enough details about the approach and evaluation to be properly evaluated at its current stage.

We are pleased that the reviewer finds the study potentially interesting and address his/her concerns below.

Some major points:

**) Did the authors check against human reads instead of the assembled reference genome?*

Unfortunately, there is no approach to directly align two sets of short reads, as such alignment is subject to a very high false positive rate due to the short but random sequence matches among reads. However, this is an interesting point and relates to whether the assembled reference genome contains “gaps” that need to be filled, as might be evident from unassembled/unmapped reads. We have approached this issue in two ways: (1) we assembled the unmapped (i.e., not previously assembled) reads and showed that these assembled transcripts are indeed part of human genome/transcriptome (by PCR validation and primate mapping), and these transcripts are missing “gaps” in the current genome assembly. (2) we took other human NGS libraries (e.g., ChIP-seq data from H3K4Me3, H3K27ac, and Input), and showed that there are human reads in those libraries that map to these newly identified “gaps”.

**) Is the complete transcript structure supported in chimp or just parts?*

Most of the newly identified transcripts are fully supported in chimp/gorilla, as 2064 of 2550 (>80%) of transcripts have >90% nucleotide match between the newly identified transcripts and chimp/gorilla genome. This suggests that some of the “gaps” in the assembly of one species might already be (assembled) in a closely related species, and thus using the assembly of closely related species is a way to fill “gaps” in the target species.

**) Are these transcripts annotated in apes and/or detected in ape RNAseq data?*

We have aligned our 2550 newly identified transcripts to all characterized and predicted genes in Chimp from Ensemble (29,160) and Genscan (44,281 transcripts) databases, which are the most comprehensive sources available and are updated based on the recent data from The Nonhuman Primate Reference Transcriptome Resource (NHPRTR). There are only 12 (of 2550) transcripts that have >50% sequence match to known chimp transcripts, of which only 1 has >90% match. This is consistent with the fact that annotation of non-primate genomes is less complete than the human genome annotation.

**) How convinced are the authors that they were able to map the complete transcript -- and vice versa, that pooling so many libraries would not lead to the combination of distinct transcripts?*

We agree that some of the newly identified transcripts may not represent full-length cDNAs, and some transcripts may be a combination of overlapping and/or alternative splicing of a gene. This is now stated on p. 15. However, given that we could not extend any of newly assembled transcripts using the unmapped reads from all cancer and normal RNAseq samples, we believe that most of the transcripts represent full cDNAs. It is difficult or impossible to determine whether the short reads are generated from distinct transcripts when the transcripts themselves overlap.

**) all results need to be put in context with control set of known lincRNA or eRNA - expression, histone modifications, length etc.*

We agree and have now compiled a list of 2098 already known lincRNAs and 17529 protein-coding transcripts and included them as a control set for expression, histone modification, and length. The results of these comparisons are incorporated into the related parts of the main text: p.7 and Supplementary Fig S3 for expression and length control, and p. 9 and Supplementary Fig S5 for histone modification control.

**) it is completely unclear how the authors mapped histone modifications -- what was the precise quantitative approach to call a modification?*

The “modification calling approach” was previously in the legend to Supplementary Table 8. We have now moved the information to a new section in the Methods labeled “Calling histone marks at genomic loci of newly assembled transcripts”. We have also quantified specificity/sensitivity of the method based on MACS peak calling.

**) if many transcripts map to putative eRNA/PROMPT sites, then the authors effectively mapped promoters and enhancers but not necessarily new genes. Most of these transcripts are evidence of transcriptionally active regions but whether they have any function is highly debated. This should be clearly discussed, eg. in the context of studies such as (Andersson et al, Nature Comm 2014)*

We thank the reviewer for bringing to our attention the paper by Andersson et al, Nature Comm 2014 discussing the exosome-mediated degradation as an approach to determine potentially functional candidates among lncRNAs. We agree that some of our newly assembled regions might represent promoter and enhancer sites rather than lncRNAs and their function(s) need to be elucidated. However, even in those cases, our assembled regions are previously missing promoter and enhancers. Moreover, 31% of these transcripts seem to have more than 1 exon (Supplementary Figure 2), more consistent with the notion of new genes. We now mention this issue in the Discussion on p. 11: “Some of these transcripts, in particular single exonic ones, might originate from promoter upstream transcripts (PROMPTs), or enhancer regions, and although they might lack immediate function (Andersson et al, 2014), the uncovered corresponding genomic sequences could harbor regulatory elements controlling gene expression.”

**) Overexpression assays should be complemented by other assays -- these RNAs are largely lowly expressed and it is hard to gain insight from this approach alone.*

We agree that the overexpression assay alone may provide only limited insight into the function of these transcripts. Our intent was to demonstrate, as a proof of concept, that these previously missed transcripts, are potentially functional and warrant further annotation. We are indeed designing shRNA knockdown experiments for these transcripts. However, knockdown of lncRNAs are known to be challenging, as pre-made shRNAs do not exist, designing new shRNAs have typically low success rate for lncRNAs, and the expression of lncRNAs are typically low. In addition, these transcripts are differentially expressed in various tissues (please refer to the response to reviewer #1), requiring the determination of the correct matching cells lines that express the transcript. Although we were unable to provide other complementary assays as the moment, we have attempted to better control our experimental setups by checking cell death, cell proliferation, differential expression in various tissues, and adding other controls beyond empty vector, as suggested by Reviewer #1. We hope future studies would determine in depth the biological function of these transcripts as well as other identified “gaps” in the current assembly of human genome.

minor points:

Fig 3A -- some expression samples appear suspicious, as they have high expression for many of the RNAs (visible as columns)

We had also noticed these samples that have high-expression for many RNAs. These belong to 3 bladder, 1 colon, and 8 lung cancer samples. We noted that 3 bladder, 1 colon and 4 of the lung samples belong to the same sequencing plate (A277), and the other 4 lung samples belong to plate (A278). It is possible that some factor causing this batch effect for these samples. Originally, we considered removing these samples, but given that these are a few samples and does not affect our results, we decided to keep them. However, we will now comment that on p. 14

Reviewer #3:

Based on GC, the majority of new assembled transcripts were removed as they had high GC content. 95% of the transcripts were assigned as lncRNAs. The authors also identified novel cancer

associated transcripts and expression correlation cis associated gene expression. The authors have also tried to show the active transcription marks with novel lncRNA expression using cell line chip seq data.

Major Issues:

1. The authors have utilized BWA, which is a DNA aligner and does not take splicing into consideration. A splicing specific aligner should have been used for the alignment.

We agree with the reviewer that the BWA does not directly take splicing into consideration. To alleviate this problem, besides the genomic DNA, we had constructed transcriptomic sequence index from GENCODEv19 comprehensive transcripts, RefSeq genes, human all mRNAs, USCS genes, Ensemble genes, lincRNAs, and human ribosomal RNA sequences and mapped against it, thus directly providing the spliced sequences to the BWA. We have now added a statement in the methods on p. 12 stating:

“...It is notable that BWA is a DNA aligner and does not directly take splicing into consideration; therefore, we included transcriptomic sequences as mentioned above to recover the reads that may cover splicing junctions.”

2. As the new human genome version hg38 is available, the authors should reanalyze the data using hg38 as reference genome.

We have indeed analyzed and mapped all the discovered transcripts against both hg19 and hg38 (statistics are in Table S3). The genomic sequences of nearly half of our newly assembled transcripts are assembled in hg38. However, the annotations of these transcripts as well as the transcripts with no genomic assembly are lacking. We now report the mapping in both hg19 and hg38.

3. The transcripts the authors claim are novel did not assemble to the genome and at least some of them could very well be variants of some other transcripts.

We agree that it is possible that some of these transcripts are alternative spliced forms of each other. However, most of the transcripts have relatively distinct and distant alignment when mapped to the chimp genome, making it unlikely to be variants of each other. Nevertheless, we added a statement clarifying this great point in the text on p. 15:

“... it is still possible that some of the newly identified transcripts may not represent the full-length cDNA and/or some of these transcripts may be a combination of overlapping and/or alternative splicing of a gene.”

4. The authors have used an in-house re-assembly program for identification of new transcripts de-novo. The authors should provide more information on how the program works and also what is the false positive frequency.

We used the ABySS (Simpson et. al, Genome Research 2009) for initial assembly, and then, iteratively merged the assembled fragments, if there were more than 5 paired-reads supporting the joining between two transcript fragments (i.e. one read mapping to one fragment and the mate mapping to another fragment).

As suggested by the reviewer, we sought to quantify the false positive rate for this process. Of 2550 novel human transcripts, 221 were obtained from joining two or more transcript fragments by our re-assembly. 176 of these 221 transcripts were assembled in hg38 genome, while other 45 were not assembled. Considering the hg38 assembly as the “gold standard”, there were 11 transcripts that were mis-joined (i.e., the alignment of the transcripts to hg38 did not contain contiguous blocks of high-identity alignments). This sets the false positive rate of our in-house re-assembly process at 6%. We modified the manuscript to clarify this point on p. 14.

5. The authors claim that new transcripts "were generally similarly or less enriched in repeats". This seems meaningless as it is difficult to assemble across repeats.

We agree that it is more challenging to assemble across repeat regions. However, we believe that, in the context, our statement (on p. 5; replicated below) compares the newly assembled transcripts with the annotated repeats across the entire human genome.

“These assembled transcripts had a median length of 404 bases (Fig 2A), and as compared to the entire human genome (Smit et al, 1996), they were generally similarly or less enriched in repeats (Fig 2B).”

6. The authors claim to have found the majority of transcripts using some simulation. It is hard to know what is missing if the full content is not presented.

Our simulation provides only an extrapolation in the context of analyzed cancer types and the parameters that were used (e.g. size of the transcripts, the complexity of sequences to be assembled, and etc.). We have now modified our statement and replicated below to include the conditions and content.

“... We next sought to estimate how many additional such transcripts in those tissues remain to be found under the same assembly conditions (e.g. length cut-off and sequence complexity). Whereas the precise number of missed transcripts is unclear, simulating the number that we would anticipate to be identified from a given number of RNA-Seq samples suggested that we have indeed found the majority of such transcripts (Supplementary Fig S2)”

7. The authors have checked the neighborhood gene of lncRNA in gorillas and chimps and then checked if those genes are expressed in humans. It is unclear how fruitful this analysis will be as the location of genes in humans can be different than gorillas/chimps.

We agree with the reviewer that the location of neighboring genes in human might be different from that of chimp/gorilla. We have now added the following on p. 16.

“... It is notable that the relative location of some genes in human might be different from that of chimp/gorilla, which could introduce noise to the prediction of lncRNAs' nearest genes in human based on the alignment to the chimp/gorilla. To potentially quantify such noise (i.e. how much the relative locations of genes in human are similar/different to that of chimp/gorilla), we obtained all the genes (~19000) in chimp/gorilla with known homology in human from <http://www.ncbi.nlm.nih.gov/homologene>. For each gene in human, chimp, and gorilla, we then found its nearest neighboring gene using “bedtools closest” program. We then checked what percentage of the nearest neighboring genes in chimp/gorilla are the same with the homologous genes and their nearest neighbors in human. We observed that 74.9% and 68.1% of relative gene neighborhoods are conserved between human and chimp or gorilla, respectively.”

8. It seems strange that you could map reads to chimp/gorilla and determine the position in the human genome via homology, but not map the reads directly to the human genome. Could you just allow more mismatches to get more alignments to the human reference sequence?

We were loose in our usage of the word “homology”. In fact, there is no direct homology between the sequence of the novel transcripts (or corresponding reads) that were mapped to the chimp/gorilla genome and the human genome. For each transcript, we predicted its potential genic neighborhood from human homologs of the chimp/gorilla genes adjacent to the transcript. Thus, none of the novel transcripts (and or their corresponding reads) could be aligned to the human genome with any aligner (Figure 2C, Supplementary Table S3).

To be more clear, we now state on p. 3: “... The sequences of these transcripts were not available (i.e., not assembled) in the human reference genome, but we could map them to the chimpanzee/gorilla genomes and then using human homologs of their adjacent genes, we could predict their relative locations in the human genome.”

9. To check the histone marks in lncRNA loci, the authors have used cell line chip seq data and aligned the data with newly identified transcripts. This is a clever idea but not sure if it is the

correct method as many of the histone marks are not on the gene body and won't align with RNA seq.

We agree that some histone marks are not typically found on the gene body; however, our choices here were limited by the available data. Moreover, H3K4me3, one of the selected marks, is known to be associated with transcription, and typically found in the promoter and is extended to the gene body. In addition, H3K27ac, which is an enhancer mark, although rare in gene body, but when present, might categorize a recently recognized class of noncoding RNAs, eRNAs. We have also provided proper controls (Supplementary Figure 5 and response to Reviewer #2).

10. In Figure 5 the authors should correlate the expression with histone marks.

Unfortunately, due to lack of histone mark data, correlating the histone marks from tissue-matching cancer cell lines and the expression obtained from patient samples is not possible at the moment. However, we have calculated the significance of observing both frequent expression and histone mark in a matching tissue using the Fisher exact test (188 transcripts \times 6 tissues; p-value $< 3E-5$; reported in Supplementary Table S9 and replicated below). This provides an indirect association between the expression and histone marks of our transcripts, similar to the left-most column of Figure 5C, which highlights 42 transcripts that had frequent expression in the tissues where the histone mark was present.

11. The authors should compare the list of novel lncRNA with other recently published articles on identification of novel lncRNAs.

Given that published articles on identification of novel lncRNAs focus on characterizing the mapped portion of the transcriptome, we do not expect overlap between our lncRNAs with those studies. However, we obtained the list of all long noncoding RNAs in human transcriptome (Iyer et al, Nature Genetics, 2015) from <http://mitranscriptome.com/>. This set includes 384066 overlapping and non-overlapping of transcripts. Using BLAT, only 60 of our novel set of 2550 transcripts had any alignment with this dataset, ranging from 5-81% of their entire length, with only 5 transcripts having $>50\%$ nucleotide match. Thus, we believe that our set is novel set of unmapped, unassembled, and unannotated portion of the human genome.

12. What is the sequence complexity of the unmapped reads? By eye, the sequences in Table S2 look OK (and the first sequence has a BLAST hit on a human BAC clone). However, it would be good to have some metric to indicate the degree to which the unmapped reads have more or less repetitive sequence than expected based upon the reference sequence. Figure S1 does address this issue, which alleviates concern that the de novo assembly mostly comes from low complexity sequences, but it is worth moving that to part of a main figure.

As suggested, we have moved this figure to Figure 2B.

13. Less emphasis should be placed on the high GC content sequences that match bacterial sequences following the de novo assembly but not in the initial screen. For example, the authors should describe these as a filter for removed sequences rather than saying "Interestingly, we observed a bimodal GC-content distribution for these transcripts".

We agree that in the context of human transcriptome, the bacterial sequences are less important. However, although these sequences had short matches to the known bacterial sequences, they were distinct from already known bacterial sequences suggesting potentially novel or uncharacterized variants of bacterial sequences. We thought this was of potential interest and worthy of comment, but we now omit the word "interestingly" and have altered the wording. If you have other suggestions on the wording, we are happy to consider them.

14. The author show some sequences align to hg38. What percentage of reads/transcripts?

Of 2550 newly identified transcripts, 1473 (57%) have an assembly in hg38 and 1260 (49%) have assembly with $>95\%$ alignment (i.e., "correct assembly") between hg38 and the transcripts. We now clarify this in the Methods on p. 17.

15. The author state removed "reads mapped to cloning vectors". Is this simply checking for reads that match the adapter sequence? Given the paper specifically focuses on unmapped reads, the author should be extra careful about possible artifacts from library preparation.

We agree that this is an important technical issue regarding obtaining unmapped reads. We used UniVec (<http://www.ncbi.nlm.nih.gov/tools/vecscreen/univec/>) the only available and comprehensive source for sequences that may be of vector origin. It contains 4383 sequences for adapters, linkers, and primers commonly used in the process of cloning cDNA or genomic DNA.

In summary, we thank the reviewers for their valuable comments and have tried to respond in detail to each point that was raised. The current manuscript has been significantly modified (e.g., Figure 2, Figure 6B-C, and methods) and we have added considerable new data in Supplementary Figures 3, 5, 6, 7, and 8, and Supplementary Table 9. We believe that the manuscript is now improved and hope it is now acceptable for publication in the journal.

2nd Editorial Decision

10 July 2015

Thank you again for submitting your work to Molecular Systems Biology. We have now heard back from the two referees who agreed to evaluate your manuscript. As you will see below, the referees are satisfied with the modifications made. As such, we are glad to inform you that the study is now suitable for publication.

Before formally accepting the manuscript we would like to ask you to address a few editorial issues.

Reviewer #1:

The authors have adequately addressed the reviewer's comments.

Reviewer #2:

I have no further specific concerns, the authors have addressed my comments.

However, on an editorial level, I am not sure this paper fits well with the scope and impact of MSB. There have been a number of similar stories over the past years; this is no worse, but as a common thing, papers that annotate all ever detected low-level expressed transcripts as generic lincRNAs, without further characterization or biological context, simply do not contribute much to our understanding.

2nd Revision - authors' response

13 July 2015

Please find enclosed our revised manuscript. We hope it is now acceptable for publication in MSB.